# Efficient FPGA Implementation of a Dual-Frequency GNSS Receiver with Robust Inter-Frequency Aiding

**DOI:** 10.3390/s21144634

**Published:** 2021-07-06

**Authors:** Kuan-Ying Huang, Jyh-Ching Juang, Yung-Fu Tsai, Chen-Tsung Lin

**Affiliations:** 1Department of Electrical Engineering, National Cheng Kung University (NCKU), Tainan 70101, Taiwan; juang@mail.ncku.edu.tw; 2National Applied Research Laboratories (NARLabs)-National Space Organization (NSPO), Hsinchu 30078, Taiwan; raymond@narlabs.org.tw (Y.-F.T.); tomlin@narlabs.org.tw (C.-T.L.)

**Keywords:** robust signal combination, multi-frequency GNSS, interference, robust statistical signal processing, FPGA receiver implementation

## Abstract

Multiple frequency global navigation satellite system (GNSS) has become more complex due to the existence of extra channels. Typically, auxiliary methods are used to synchronize the second signals at other bands by aiding the acquired channel parameters. However, there are critical limitations because the reception of GNSS signals is subject to uncertainties due to noise carrier injection or circuit interference. The relationship between the two Doppler frequencies can be affected by uncertainties. Therefore, we aimed to implement an efficient dual-frequency field-programmable gate array (FPGA), performing a direct aid tracking method for the secondary channel to achieve resource efficiency and inner aid robustness. A robust estimator that directly links two loops in the two bands is proposed. In this scheme, (1) a robust estimator able to cope with uncertainty; (2) a primary tracking scheme to obtain the error boundary, and (3) a tracked bit-boundary for the initial code phase of the second channel are used. Based on experiments on the FPGA, the robust channel link can achieve direct aid tracking, and 31.02% of the original hardware resources from the aided acquisition module were released satisfactorily.

## 1. Introduction

Modernized GPS is transmitting two new signals on the L2 and L5 bands: the L2 civil (L2C) signal is transmitted at 1227.60 MHz, and the L5 signal is transmitted at 1176.45 MHz, both modulated with their own spreading code. Multi-frequency receivers led to the development of new processing techniques, which assure higher-value commercial applications. For example, a generalized multi-frequency carrier tracking architecture is proposed in [1], which can cope with frequency-selective fading; multi-frequency real-time kinematic (RTK) and precise point positioning (PPP) receivers achieve centimeter-level positioning accuracy [2,3].

Nowadays, many dual-frequency GNSS receivers use “assisted acquisition” to acquire the signal at the secondary band, which conducts an assist procedure between the L1, L2, and L5 acquisitions. The studies in [4,5,6] synchronized the signals at the second band by exploiting the channel parameters using the primary acquisition. Nonetheless, if the jitter frequency ratio and outlier occur in the auxiliary information, the performance degradation of these channel assistances is inevitable such as increased computational load and decreased robustness due to the actual observed frequency in the second band is different from the original relationship derived in [6]. Additionally, the characteristics of diverse multi-band spreading codes may prevent the primary acquisition in the first band to point to the correct code phase of the second band within the “one-shot” window. This could be due to different code period durations. For example, in a L1/L2 dual-frequency receiver, the 20 hypothesis tests must be checked in [4,7], which indicates that an aided acquisition of the secondary channel must be applied to check each candidate of the secondary code phase. However, the proposed method further considers the primary tracking scheme with a new code phase alignment strategy using the tracked bit boundary, which takes the tracked navigation edge from the first channel to point to the first chip of the secondary code phase. In this case, a direct aid-to-track link for the secondary (slave) channel can be performed to achieve a higher efficiency, due to the “ambiguity” mentioned in many lectures [6,8], which can be addressed by such bit-boundary alignment in the primary tracking stage. Therefore, the massive amount of logic originally used in typical signal aided acquisition in the secondary channel can be released with satisfaction.

To verify these situations, we designed and analyzed a dual-frequency GNSS radio frequency front end (RFFE) board, implementing an efficient dual-frequency GNSS FPGA receiver with the proposed robust inter-frequency aid. In contrast to typical aid acquisition, it creates a robust interaction between a primary tracker and the tracker in the second band, ensuring an intrinsic relationship using a primary tracking error bound. When the master tracking loop is in the lock state, the link between primary and secondary channels is exploited to initialize the secondary tracking loop, avoiding a second acquisition process.

In the robust estimation for linking the two loops, the structural uncertainty of the models is applied and divided into two main categories: random and deterministically unknown [9,10,11]. For example, the standard errors in variables (EIV) model is a deterministic unknown matrix, where the estimate is based on the noisy observations of this matrix [12,13]. As perturbation separates the estimates from the true relationship for these aiding techniques, two bands are linked using calibration and a robust linker. A model that estimates unknown deterministic error parameters in a linear system with an error in structure is applied in the estimator. In order to introduce robustness, the unknown-but-bounded (UBB) uncertainty framework [14,15,16] is adopted. In these frameworks, estimators are designed to minimize the worst case over all bounded data vectors, and the estimator can induce robustness by this unknown-but-bounded (UBB) uncertainty, maintaining the robustness to meet the “one-shot” requirement and holding the link.

We implemented the robust inner aid scheme on a Xilinx All-Programmable FPGA [17]. In Section 4, architecture and implementation efficiency are discussed. In addition, a criterion about optimizing the loops is proposed; the resources and computational load are analyzed. Then, a verification environment is setup in Section 5, performing the on-chip evaluations. This showed that the performance was almost the same when $C/N_{0}>35\;\mathrm{db}-\mathrm{Hz}$. However, when the signal-to-noise ratio (SNR) was low, the robust inner aid could maintain the link between two loops in two bands.

## 2. Synchronization of Multi-Band GNSS

In all GNSS, the receiver must achieve three levels of synchronization—the code phase, carrier, and the data bit—through the acquisition and tracking loop in each band. A combination of the GPS L1 gold code and the L2 CM code was considered, which has the characteristics of different code periods. After the front-end processes, received GNSS signals, $r_{L1}(nT_{s})$ and $r_{L2}(nT_{s})$ samples are down-converted to the intermediate frequency (IF) and expressed as:(1)$$\begin{array}{l} r_{L1}(nT_{s})=P_{L1}d_{}(nT_{s}-\tau_{L1})c_{L1,C/A}(nT_{s}-\tau_{L1})\\ \cdot \exp\left(2\pi j(f_{IF,L1}+f_{D,L1})nT_{s}+j\varphi_{L1}\right)+n_{L1}(nT_{s})\\ r_{L2}(nT_{s})=P_{L2}\exp\left(2\pi j(f_{IF,L2}+f_{D,L2})nT_{s}+j\varphi_{L2}\right)\\ \cdot \left[d\;(nT_{s}-\tau_{L2})c_{L2,CM}(nT_{s}-\tau_{L2})+c_{L2,CL}(nT_{s}-\tau_{L2}+kL)\right]+n_{L2}(nT_{s}) \end{array}$$
where $T_{s}$ is the sampling rate; n is the sample; $P_{L1},P_{L2}$ are the received power; d=±1 presents the data symbols; $c_{L1,C/A}$, $c_{L2,CM}$, and $c_{L2,CL}$ are the received codes with phases $\tau_{L2}$, $\tau_{L1}$. $f_{IF,L2}$, respectively, and $f_{IF,L1}$ represent the IF together with received Doppler frequency, and $f_{D,L1}$ and $f_{D,L2}$, L refer to the samples in the civil moderate (CM) code period, k, in the interval 1 < k<75 interlacing the civil long (CL) code. $\varphi_{L1}$ and $\varphi_{L2}$ are the received carrier phase in radians, and $n_{L1}$ and $n_{L2}$ are the zero-mean white Gaussian noises.

The acquisition block computes the signal cross-ambiguity function (CAF) [18], which can be expressed as:(2)$$\begin{array}{l} Z\left(\hat{\tau },{\hat{f}}_{D}\right)=\left|\sum_{n=0}^{N-1}r_{}(nT_{s})\cdot c_{l}(nT_{s}-\hat{\tau })\cdot \exp(-j2\pi {\hat{f}}_{D}nT_{s})\right|\\ =P\left|\sum_{n=0}^{N-1}c_{r}(nT_{s}-\tau )\cdot c_{l}(nT_{s}-\hat{\tau })\cdot \exp(-j2\pi (f_{D}-{\hat{f}}_{D})nT_{s}+j\varphi )\right| \end{array}$$
where P is the received signal power; $r_{}(nT_{s})$ represents the received signal; $c_{r}$ is the received code; $c_{l}$ is the local code replica; φ represents carrier phase of the received signal; N is the number of all samples in a coherent integration time. The acquisitions provide minimized residual errors of the code phase and Doppler frequency, $\Delta \tau =\tau_{D}-{\hat{\tau }}_{D}$ and $\Delta f_{D}=f_{D}-{\hat{f}}_{D}$, to maximize the CAF.

### 2.1. Conventional Aid Acquisition Approaches

In modernized GNSS, long-code spread in new bands may be resource-demanding and time-consuming in terms of acquisition, as extra bands must still use extra resources to achieve synchronization. Conventional aid acquisition approaches can potentially reduce the search region and computational load as demonstrated in [4,7], where the channel parameters, code phase Doppler frequency, at frequency $f_{1}$ are acquired to speed up the signal acquisition from the secondary frequency. In particular, two received Doppler frequencies are related by the two carrier frequencies [4,8], defined as $f_{1}$ and $f_{2}$, expressed as:(3)$$f_{D,L1}=\frac{f_{1}}{f_{2}}f_{D,L2}$$

This is briefly illustrated in Figure 1a. However, some limitations to these schemes are:

The primary acquisition scheme is time-consuming, its assist rate is lower than the primary tracking scheme, which updates the auxiliary information each millisecond.If an outlier exists in the auxiliary information, a performance degradation occurs.

### 2.2. The Concept of the Proposed Robust Inner Frequency Aid

As demonstrated in Figure 1b, the aided tracking for the signal in the second band is realized. In this concept, once the second tracking loop is in a lock state, the resources and computational load of the assisted acquisition in the second band is released since direct tracking is achieved. To achieve this goal, the method must point out the precise code phase and the Doppler frequency in the second channel with robust solutions, or the method will fail because the loop in the second band cannot tolerate the initial tracking error, leading to loss of locks and invalid aid synchronization.

#### 2.2.1. Limitation of Robustness

Various errors such as thermal errors may perturb the frequency relationship of (3). Specifically, two received Doppler frequencies $f_{D,L1}$ and $f_{D,L2}$ may be incorrectly related because of a jittery frequency. For example, when ionospheric scintillation or circuit mismatch occurs, increased noise appears in the two independent channels, leading to additional uncertainty. Once the relationship between two channel parameters is affected, the conventional link may be invalid.

To clarify the actual error, as can be seen in Figure 2, we implemented two dual-frequency radio frequency front ends (RFFEs) using a FPGA to collect outputted IFs for error analysis. The radio frequency integrated circuits (RFICs) of the chipsets MAX2112 and MAX2769 were equipped to process the RF signals at the L1 and L2 bands, which include the low noise amplifier (LNA) and radio frequency variable-gain amplifiers, I/Q analog down-converting mixers and bandpass filters with cutoff frequency controllers. Because the total cascaded noise figures of the RFICs were computed as low as 1.4 dB, the error of the local delta-sigma fractional-N frequency synthesizer in each RFIC was +40 Hz ~ −40 Hz. Therefore, the relationship uncertainty of the Doppler frequency between two obtained IFs occurred. Therefore, the baseband processor has to build a robust signal processing strategy to deal with these uncertainties for a better performance.

To facilitate design of the robust strategy, a parameter fine tune flow, which contains a rough acquisition and post-processed parameter fine tune framework, was applied to find the precise Doppler frequencies and code delays between two channels. First, the coarse acquisition uses a 500 Hz search step to fast search a grid that contains the true signal, which determines any correlator output greater than the detection threshold as a detected result. Then, the post-processing fine tune framework spends a lot of time to finely adjust the local parameter generator to approach the absolute maximum of the correlator output in this detected grid. In addition, the DFT zero padding, which adds zeros to the end of a time-domain signal for obtaining a higher DFT resolution, was used to achieve a better fine-tune result.

As a consequence, as can be seen in Figure 3, we can thereby observe that the accurate experimental relationship between the two channels is separated from the theoretical relationship by the bias.

#### 2.2.2. Limitation of Different Code Periods from Two Bands and Corresponding Process

When L1 aids the L5 GNSS receiver, the C/A code tracking on the L1 band is used to aid the processing on L5, with a periodic code of 1023 chips and a chipping rate of 1.023 Mcps. The L5 signal is characterized by a 10,230 code, which is ten times longer than L1 but the code rate is ten times faster than L1, hence, the code period remains the same. In this condition, the primary code delay directly points to the code phase in the second band, where the relationship of the two received code phases $\left(\tau_{L1},\tau_{L5}\right)$ can be derived as:(4)$$\tau_{L5}=10\times \left(\tau_{L1}-\Delta \tau_{Ion,L1-L5}\right)+w_{{{}_{L1-}}_{L5}}$$
where $w_{{{}_{L1-}}_{L5}}$ is the code error in this primary-assisted equation; $\Delta \tau_{Ion,L1-L5}$ represents the delay difference between two code phases, caused by the ionosphere error of between two channels depending on the total electron content (TEC), where the unit of the TEC is $10^{16}{\;\mathrm{electrons}/m}^{2}$. To verify, we used the GSS-8000 simulator, which refers to the Klobuchar ionospheric model used to generate the ionospheric delay in RF signals [19,20]. Based on the experiments in Section 5.1, the obtained error code delay of $\Delta \tau_{Ion,L1-L5}$ was within 0.03 chip, which was smaller than the initial error tolerance of the slave loop (0.5 chip). Hence, the impact of the ionosphere error can be overlooked in the initialization process of the assisted slave loop as accuracy of the assisted data, $\tau_{L1}$, from the primary channel is the main concern.

Nevertheless, for a dual-frequency receiver, different combinations may be selected. In the L2 channel, the Return-to-Zero (RZ) is often used for decomposition [4]. Specifically, the RZ CM method replaces CL chips in the interleaving L2C code structure with zeros in (1), then the local RZ CM code alternated between CM chips and zeros with 20 ms code duration. The L2 RZ CM code was 20 times longer than the L1 C/A code, and Figure 4 shows that ambiguity may appear under combinations with different code periods such as $amb_{CM}$ and $amb_{CL}$. Therefore, a relationship of the two code phases $\left(\tau_{L1},\tau_{L2}\right)$ can be derived as:(5)$$\begin{array}{l} \tau_{L2,CM}=\left(amb_{CM}-1\right)\times c_{L1}+\tau_{L1}-\Delta \tau_{Ion,L1-L2}+w_{{{}_{L1-}}_{L2}},amb_{CM}\in \left[1,2,\ldots ,20\right]\\ \tau_{L2,CL}=\left(amb_{CM}-1\right)\times c_{L1}+\left(amb_{CL}-1\right)\times c_{L2,CM}+\tau_{L1}-\Delta \tau_{Ion,L1-L2}+w_{{{}_{L1-}}_{L2}},amb_{CL}\in \left[1,2,\ldots ,75\right] \end{array}$$
where $\tau_{L2,CM}$ is the RZ CM code; $\tau_{L2,CL}$ is the RZ CL code; $w_{{{}_{L1-}}_{L2}}$ is the code error in this primary-assisted code estimation; $c_{L1}$ is the L1 code duration; $c_{L2,CM}$ is the L2 RZ CM code duration; and $amb_{CL}$ is in the interval 1<k<75 interlacing the CL code. $\Delta \tau_{Ion,L1-L2}$ represents the difference in the ionosphere delay error between two channels. In the assist Equation (5), it was assumed that the L1 C/A code and the L2CM code had the same code rate.

The ionospheric delay of each channel is solved as $\mathrm{TEC}\times \left(40.3\;/cf_{}^{2}\right)$ using the Klobuchar algorithm [21,22], which is a curve fitting technique that considers eight inputted Klobuchar coefficients obtained from the navigation messages. This delay was found to be proportional to the TEC and inversely proportional to a square of the carrier frequency f and speed of light c. In this model, the value of 40.3 was the so-call Klobuchar factor [20], which is computed to reduce the ionospheric root mean square range error by 57% [23]. Typical receivers use the aided acquisition in the L2 channel to find the ambiguity to solve ${\hat{\tau }}_{L2}$ in Equation (5). To reduce time consumption in detecting the ambiguity, the bit-boundary assisted signal alignment is used due to the rising edge of the navigation bits being simultaneously modulated at the same time in the transmitter. As can be seen in Figure 5, the primary tracked bit edge can point to the first chip of the L2CM code. Therefore, the phase of the L2CM code can be determined within the “one-shot” estimation. That is to say, the $amb_{CM}$ is solved as “1” for the assist Equation (5) by the tracked bit edge. Hence, a simplified relationship of the two code phases $\left(\tau_{L1},\tau_{L2}\right)$ is shown as:(6)$$\tau_{L2,CM}=\left(\;1\;-1\;\right)\times c_{L1}+\tau_{L1}-\Delta \tau_{Ion,L1-L2}+w_{{{}_{L1-}}_{L2}}$$

In addition, when the signal-to-noise ratio becomes low or noise becomes strong, the tracked bit edge from the primary loop in the first channel could be a better robust indicator. This kind of indicator is verified in Section 5.3.

In general, the navigation bit period is longer than the code period. However, this has a limitation when a code period is greater than the navigation bit period such as using the tracked bit boundary to assign the $\tau_{L2,CL}$. In this case, the ambiguity of the RZ CL code, $amb_{CL}$, cannot be solved by the bit-boundary given its extremely long code duration (1500 ms).

## 3. Proposed Aided Tracking and Robust Inner Aiding Scheme

The proposed processing scheme is represented in Figure 6. The main operations implemented are as follows:

The tracking results from the primary channel are used to render the upper-bound of the error-assisted data. The uncertain regions can be propagated from the locked tracker in the second band, as described in Section 3.2.The noise statistics of the primary channel are estimated and used to form a robust least squares (RLS) that guarantees the robustness of the relationship. Specifically, this RLS considers a modified dual-frequency GNSS model.The bit-boundary is tracked by the primary loop. Therefore, the code phases in the L2 and L5 bands can thereby be assigned for a fast decision.

### 3.1. Robust Link between Two GNSS Bands

In this scheme, a deterministic non-uniform robust estimator [14,15,16], which receives unknown-but-bounded parameters associated with statistical signal processing, is presented to obtain a robust link between two loops.

#### 3.1.1. The Extended Model with Structural Uncertainty

In order to address uncertainties and deviations in the frequency Doppler ratio, an extended model is adopted:(7)$$f_{D,\;L1}+n=\left(\frac{f_{1}}{f_{2}}(1+s)\right)\;f_{D,\;L2}+b_{f}$$
where s is a scale factor; $b_{f}$ is a bias term; and n is a residual noise term, which is assumed to be zero mean and Gaussian. Next, an EIV model, which corresponds to Equation (7) and describes the situations when errors result from two channels, is expressed as:(8)y=(H+ΔH)X+B
where $y={\left[f_{D,\;L1}\;\Delta \;f_{D,\;L1}\;\Delta \tau_{L1}\right]}^{T}$ is the measurement obtained from the primary tracking loop (see Figure 6). y  is composed of $y^{†}+e$, which contains the true channel parameters $y^{†}$ and zero-mean Gaussian noise vector e with a variance of $\sigma_{e}^{2}$. $X={\left[f_{D,\;L2}\;\Delta \;f_{D,\;L2}\Delta \tau_{L2}\right]}^{T}$ is the true received channel parameters; ${\hat{X}}_{RLS}={\left[{\hat{f}}_{D,\;L2}\;\Delta \;{\hat{f}}_{D,\;L2}\Delta {\hat{\tau }}_{L2}\right]}^{T}$ is the robust estimate that provides the auxiliary Doppler frequency, Doppler frequency drift, and the code phase drift to aid the slave tracking loop. H is an observation matrix, which represents the frequency transformation of $f_{1}/f_{2}$ between the two bands, expressed as:(9)$$H=\left[\begin{array}{ccc} \frac{f_{1}}{f_{2}} & \frac{f_{1}}{f_{2}} & 0\\ 0 & \frac{f_{1}}{f_{2}} & 0\\ 0 & 0 & 1 \end{array}\right]$$

ΔH represents an additional uncertainty of H, expressed as
(10)$$\Delta H=\left[\begin{array}{ccc} \frac{f_{1}}{f_{2}}s & \frac{f_{1}}{f_{2}}s & 0\\ 0 & \frac{f_{1}}{f_{2}}s & 0\\ 0 & 0 & 0 \end{array}\right]$$

Due to the thermal carrier injection or clock mismatch, the bias matrix $B={\left[b_{f}\;0\;0\right]}^{T}$ is added as a column vector of the Doppler frequency bias in the EIV model. In Section 3.1.3, a specific mitigation of the fixed bias $b_{f}$ is further performed to make the cell search more robust toward a wide range of initial frequency offsets.

#### 3.1.2. Robust Estimation

The RLS estimator is used to guarantee robustness of the inter-frequency aiding. As long as an upper bound of the structural uncertainty is caught, the robustness of the estimates can be obtained. The upper bound of structural uncertainty is defined as $\Lambda_{H}$, so that ${\left\|\Delta H\right\|}_{2}\leq \Lambda_{H}$, which is analyzed in Section 3.2. To solve the robust estimator with the EIV model in Equation (8), the robust minimax principle gives an expression to the ${\hat{X}}_{RLS}$ as follows:(11)$${\hat{X}}_{RLS}=\underset{{\hat{X}}_{RLS}}{\min}\left\{\underset{{\left\|\Delta H\right\|}_{2}\leq \Lambda_{H}}{\max}{\left\|\left(H+\Delta H\right){\hat{X}}_{RLS}-y+B\right\|}_{2}\right\}$$
the min–max robustness problem can be arranged as Equation (12), which gives an optimized solution under the upper bound of the structural uncertainty.
(12)$${\hat{X}}_{RLS}=\underset{{\left\|\Delta H\right\|}_{2}\leq \Lambda_{H}}{\min}\left\{{\left\|H{\hat{X}}_{RLS}-y+B\right\|}_{2}+\Lambda_{H}\times {\left\|{\hat{X}}_{RLS}\right\|}_{2}\right\}$$

By deriving a minimization of Equation (12), the robust solution can be expressed as
(13)$${\hat{X}}_{RLS}=\left[{\left(H^{T}H+\mu I\right)}^{-1}H^{T}\right](y-B)$$
where $\mu =\frac{\Lambda_{H}{\left\|H{\hat{X}}_{RLS}-y+B\right\|}_{2}}{{\left\|{\hat{X}}_{RLS}\right\|}_{2}}$ is a regularization parameter that damps the propagated error in the calculated approximation of ${\hat{X}}_{RLS}$. One of the popular methods for determining optimal μ is the generalized cross validation (GVC) [24,25,26]; these methods find μ by evaluating the impact of an influence matrix in the robust estimation, defined as $H\left[{\left(H^{T}H+\mu I\right)}^{-1}H^{T}\right]$ as it minimizes the expected value of $E\left[{\left\|y-H{\hat{X}}_{RLS}\right\|}_{2}\right]$. On the other hand, when the upper error bound of the measurements and structural uncertainty is obtained, Refs. [14,15] discussed several bounding conditions and then used a singular value decomposition (SVD) of H to perform a predictive risk optimization, establishing an μ choice rule.

To solve ${\hat{X}}_{RLS}$, the SVD of H is taken to establish a realized cost function in Equation (18) to tune the optimized μ. The SVD of H is addressed as
(14)$$H=U\Sigma V^{T}$$
the orthogonal U and V matrices are decomposed as:(15)$$U=\left[\begin{array}{ccc} 0.8507 & 0 & -0.5257\\ 0.5257 & 0 & 0.8507\\ 0 & 1 & 0 \end{array}\right]\;,\;V=\left[\begin{array}{ccc} 0.5257 & 0 & -0.8507\\ 0.8507 & 0 & 0.5257\\ 0 & 1 & 0 \end{array}\right]$$
and $\Sigma =diag\left[\begin{array}{ccc} 2.0765 & 1 & \;0.7931 \end{array}\right]$ is a diagonal matrix with nonnegative diagonal entries. These entries are singular values of H. With the SVD, the squared Euclidean norms of ${\left\|H{\hat{X}}_{RLS}-y+B\right\|}_{2}$ and ${\left\|{\hat{X}}_{RLS}\right\|}_{2}$, which are the denominator and numerator of μ, can be decomposed (see Appendix B). Therefore, μ can be reformed as:(16)$$\mu =\Lambda_{H}\frac{\left(\mu {\left\|{\left(\Sigma^{T}\Sigma +\mu I\right)}^{-1}U^{T}(y-B)\right\|}_{2}\right)}{{\left\|\Sigma {\left(\Sigma^{T}\Sigma +\mu I\right)}^{-1}U^{T}(y-B)\right\|}_{2}}$$
by arranging Equation (16) as
(17)$$\mu {\left\|\Sigma {\left(\Sigma^{T}\Sigma +\mu I\right)}^{-1}U^{T}(y-B)\right\|}_{2}-\Lambda_{H}\times \left(\mu {\left\|{\left(\Sigma^{T}\Sigma +\mu I\right)}^{-1}U^{T}(y-B)\right\|}_{2}\right)=0$$
a cost function is obtained in function of μ, and expressed as
(18)$$J\left(\mu \right)={\left(U^{T}y-U^{T}B\right)}^{T}\left(\Sigma^{T}\Sigma -\Lambda_{H}I\right){\left(\Sigma^{T}\Sigma +\mu I\right)}^{-1}{\left(\Sigma^{T}\Sigma +\mu I\right)}^{-1}U^{T}\left(y-B\right)$$

Therefore, for a specific $\Lambda_{H}$, by using the approximation such as the Newton-Raphson method and bisection search, the optimized μ can be found, so that J(μ)≈0.

For such a robust estimator, internal errors are not expected in the inner aid from the master to slave loop. Thus, the Gauss-Markov theorem [27] was applied to ensure the unbiased characteristics of the RLS (see Appendix C).

#### 3.1.3. Bias Calibration

The conventional aid scheme does not have any preprocessing to the $b_{f}$, hence, the failure link may occur when a wide range of initial frequency offsets appear between two channels. In this technique, the mitigation of the fixed bias $b_{f}$ is taken to make the cell search more robust toward a wide range of initial frequency offsets from environmental errors such as the clock sensitivity or mismatch.

First, tracked bit edges provide a robust initial code phase and its drift for the secondary channel. Using such a pre-linked robust auxiliary code phase, the second channel can wipe off its spreading code by a mixer. Then, a DFT can obtain a rough Doppler frequency $b_{DFT}$ for calibrating $b_{f}$ between two channels in the RLS estimator, expressed as:(19)$${\hat{b}}_{f}=f_{D,\;L1}-b_{DFT}$$

It is worth noting that it does not need to perform the calibration (19) for each satellite, assigned with different pseudo-random noise (PRN) codes. As can be seen in Figure 3, the receiver executes exactly only once, the code pre-linked, mixer, and DFT against one of the PRN codes for solving the fixed bias, then, all satellites with every different PRN codes no longer need to perform the DFT due to initial frequency offsets ${\hat{b}}_{f}$ between two bands have been processed.

As a consequence, with the obtained calibrated factor ${\hat{b}}_{f}$, we can use the continuous robust aid scheme with the EIV model (13) to immediately solve all Doppler frequencies of all satellites in the L2 band, describing all channel variation by the primary tracking scheme with a higher assist rate (1 aid per ms). The specific reasons why the robust link is a better scheme than using $b_{DFT}$ to initial the slave band were analyzed as:

A real-time receiver takes a huge amount of execution time to perform such precision zero-padding long DFT (30 ms) to obtain one $b_{DFT}$, thus, a PLL initialization of the slave loop bears too huge a drift frequency range given mobility of the satellites.In contrast, the robust link can take ${\hat{b}}_{f}$ using the EIV model, which considers the jitter ratio, to rapidly estimate all channel parameters of all PRNs with high assist rate (1 aid per ms), accuracy (20 Hz), and robustness, better initializing the slave loop.

### 3.2. Criterion for Confirming the Direct Aid Tracking by a Boundary Analysis

In this section, we provide a criterion for confirming whether the primary tracking loop can directly drive the slave loop or not. First, a transformation proposed by [1] was introduced to allow the standard deviation of the error in the second band can be estimated by the ratio of $f_{1}/f_{2}$. The standard deviation of the second band can be estimated and expressed as:(20)$${\hat{\sigma }}_{L2-L1}=\frac{f_{2}}{f_{1}}\times \sigma_{L1}$$
where $\sigma_{assisT,L2-L1}$ is the error of standard deviation of the aided data for the second band. Based on this rule, we further considered a situation when channel perturbation occurs using a modified transformation, which involves a jitter ratio of $\left(1+s\right)\frac{f_{2}}{f_{1}}$ with calibrated factor s for better modeling, expressed as:(21)$${\hat{\sigma }}_{L2-L1}=\left(1+s\right)\frac{f_{2}}{f_{1}}\times \sigma_{L1}$$

Focusing on the purpose of this section, which desires to find an upper error bound of the aided data to the second channel for ensuring the direct aid tracking. The bound of the jittery factor is represented as $s_{b}$.

To find the $s_{b}$, a long-term statistical test (>1 h) was performed against the same RFFE. For example, we used the post-processing analysis to find the long-term statistical standard deviation of the two channels. Using the GSS-8000 simulator in Section 5, we performed experiments to find ${\hat{s}}_{b}$, which corresponded to each obtained tracked signal-to-noise ratio of the primary tracking loop. In this manner, the ${\hat{s}}_{b}$ is obtained as:(22)$${\hat{s}}_{b}=\left(\frac{f_{1}}{f_{2}}\times \frac{3\sigma^{†}{}_{L2}}{3\sigma^{†}{}_{L1}}\right)-1$$
where $3\sigma^{†}{}_{L2}$ and $3\sigma^{†}{}_{L1}$ are the 3-standard deviations upper bound of the two channels, obtained from the long-term statistical test. Thus, the $3\sigma_{L2-L1}$ can be estimated as
(23)$$3{\hat{\sigma }}_{L2-L1}=\left(1+s_{b}\right)\frac{f_{2}}{f_{1}}\times 3\sigma_{L1}$$
where $3\sigma_{L1}$ is the tracked error bound of the primary loop in the first channel, which is typically less than 30 Hz according to the specification of a locking condition.

As a consequence, confirming whether the primary tracking loop can directly drive the slave loop or not can be achieved by evaluating the following rule:(24)$${\left\|3\sigma_{L2-L1}\right\|}_{2}\leq e_{r}$$
where $e_{r}$ is the initial error tolerance of the slave loop. Therefore, based on this criterion, we can design a primary loop with a small enough $\sigma_{L1}$ in Equation (23) to satisfy the inequality for ensuring the direct aid tracking.

## 4. Efficient FPGA Implementation of a Dual-Frequency Receiver with the Robust Aid

In this section, as shown in Figure 7, we discuss an efficient implementation of a dual-frequency GNSS FPGA equipped with the on-chip robust aid. First, the dual-frequency GNSS architecture on the FPGA is briefly introduced. Next, algorithm partition is discussed. In addition, a criterion about implementing the slave loop and master is proposed to optimize the robust inter-frequency aid. Finally, the computational load is analyzed.

### 4.1. Implementation of GNSS FPGA

In this section, we focused on the FPGA architecture, and implemented blocks, which contain the proposed robust inter-frequency aiding scheme.

#### 4.1.1. Basic FPGA Architecture and On-Chip Hardware Units

The proposed scheme was implemented on the Xilinx Zynq-7000 FPGAs [17], which integrates a processing system (PS) based on an ARM Cortex-A9 CPU and the FPGA programing logic (PL) into a single system-on-a-chip (SoC), programed in the hardware description language (HDL). These two cores are connected by the advanced microcontroller bus architecture (AMBA) for a large-bandwidth internal data transmission. The FPGA includes an array of configurable logic blocks (CLBs) such as customizable logic gates, flip flops, and logic wires. These CLBs are surrounded by input/outputs (I/Os), which can communicate to the external devices. Two block RAMs (BRAMs) are equipped on the FPGA for a low-latency data access. A large-capacity double data rate synchronous dynamic RAM (DDR) is connected to store the long-term I/F data (>1 G bytes).

#### 4.1.2. On-Chip Implemented Blocks and Operation Process of the GNSS FPGA

Based on this FPGA architecture, the implemented blocks in Figure 8 are discussed. First, an embedded bootloader boots up the embedded CPU by bringing in the real-time operating system (RTOS) kernel, using a trigger comment to reset the whole FPGA. Then, the dual frequency IF data from the RFFEs are buffered by on-chip SRAMs. In the primary channel, an acquisition block searches the coarse detected results; a costa loop initializer drives the parallel primary tracking block. Next, the robust aid is executed by the double precision floating-point unit (FPU) in the CPU. Finally, the double FPU forms the pseudo ranges for the positioning block.

### 4.2. Efficient FPGA Implementation Strategies for the Robust Inner Aid

In this section, we discuss the efficiency issues of the FPGA implementation. Based on the mentioned architecture, design strategies are proposed.

#### 4.2.1. Trade-Off Strategy against Speed, Power, and Resources of the GNSS FPGA Implementation

To achieve an efficient FPGA implementation, a pipeline-parallel method is applied to trade off the speed, power, and resources [28]. As shown in Figure 9, a pipeline strategy was used to divide the master acquisition into multiple stages to reduce the instruction latency; a parallel architecture was built to process the multi-tracking channels. Considering the utilization efficiency of the CLBs, the resource consuming DFT was executed in a time-division multiplexing (TDM) scheme. Specifically, only one DFT module was equipped in the search block to sequentially perform the DFT for all channels. In addition, the clock gating scheme was used to reduce dynamic power dissipation by removing the clock signal when the module is not in use, so that the flip-flops in the gated modules do not have the power-consuming low-to-high switch. Hence, 43% dynamic power can be saved. Under this on-chip setting, the worst case of the time to first fix was verified as 25 s in cold start tests.

#### 4.2.2. Efficient Tasks Partitioning Scheme for the Heterogeneous Core of the GNSS FPGA

A suitable partition for algorithms of a dual-frequency receiver on the Programmable Logic (PL) or CPU Processing System (PS) is key to the computational speed and accuracy, such as [29]. As can be seen in Table 1, tasks of the GNSS signal receiving are separately dispatched. The FPGA PL is used to accelerate the GNSS signal processing. On the other side, some units that require double precision floating-point calculations are computed by the double FPU on the embedded CPU, such as the discriminator and positioning. The Free-Real-time Operating System (Free-RTOS) is transplanted into the embedded CPU as a multi-thread scheduler, which efficiently orders the execution priority of all modules.

#### 4.2.3. Efficient Combination of the Slave and Master Loops for Inner-Frequency Aid

As for properties of the primary and slave loops, to reduce the computational load and improve accuracy of the auxiliary information, the primary loop can have characteristics of fast response and zero steady-state error, which allow the RLS estimator to rapidly obtain the unbiased measurements y.

Based on the criterion (24), the slave loop can be designed to have a high initial error tolerance $e_{r}$. Therefore, the inner-frequency aiding scheme can be efficiently performed. We defined a 1% admissible tracking error of the primary tracking loop. Then, as shown in Figure 10, an appropriate time to obtain the measurement for the inner-frequency aiding process was defined as the settling time of the primary loop, where the settling time can be obtained by the Monte Carlo simulation for different receiver dynamics, the results of which are further discussed in Section 5.2.

### 4.3. Computational Load Analysis of the Aiding Schemes on the FPGA

In this section, the time spent to achieve synchronization in the slave loop is defined as the computational load. When using the conventional aid acquisition scheme, as the typical slave acquisition cannot start until the assist data are obtained from the primary acquisition, its computational load was thereby deduced as the sum of (1) a running time $R_{E\;}$ taken from typical channel estimation using (3); (2) a mean acquisition time (MAT) of the acquisition in the first (primary) channel; and (3) MAT of the acquisition in the secondary channel, defined as $MAT_{SlV}$, whereas the computational load of the primary tracking scheme considers a sum of (1) a running time $R_{RLS\;}$ taken from performing the RLS link (13); (2) MAT of the acquisition in the first (primary) channel, $MAT_{Pri}$; and (3) settling time of the first (primary) channel, $T_{S{,}_{}pri}$. In summary, the computational load of the conventional aid acquisition $T_{C}$ and the proposed primary tracking $T_{P}$ were analyzed as:(25)$$\begin{array}{l} T_{C}=MAT_{Pri}+MAT_{Slv}+R_{E\;}\\ T_{P}=MAT_{Pri}+T_{S,\;}{}_{Pri}+R_{RLS\;} \end{array}$$

If $T_{C}>T_{P}$, it indicates that the primary tracking scheme had a better performance than the typical method as typical methods have to endure a heavier computational load. On the other hand, the mean acquisition time (MAT) was addressed under different SNRs with different probabilities of detection, expressed as follows [30]:(26)$$MAT=\frac{2-P_{D}}{P_{D}}(M-1)T_{P}(1+P_{F})+\frac{T_{P}}{P_{D}}$$
where *M* is the number of hypothesis tests on a specified searched plane and $T_{P}$ (ms) is the time consumed by the FPGA to process each grid. $P_{D}$ is the probability of detection and $P_{F}$ is the probability of false alarm, obtained by the following experimental setting.

## 5. Experiments and Results

In the previous section, the efficient implementation of a dual-frequency GNSS FPGA with the robust inner aid are discussed. In this section, the experiments, setup, computational load as well as the performance of the robust aid are demonstrated.

### 5.1. Setup for Experiments and Verifications

As can be seen in Figure 11, the mentioned RFFE was connected to the GNSS FPGA, the 2-bits I/Q data were rated at 16.368 MHz, processed by the implemented FPGA in real-time. A GSS-8000 SPIRENT GNSS simulator was applied to generate RF signals that covered several test cases with different signal-to-noise ratios. A high-precision rubidium atomic clock was used as an extra external clock source and inputted to an independent counting module on the FPGA as a rubidium atomic clock can accurately count the processing time of the implemented blocks. An oscilloscope was involved to record the on-chip signals and measure the voltage. The on-board DDR3 connected to the FPGA can record the long-term massive IF data.

### 5.2. Evaluate the Time Spent on Synchronization by the FPGA

In general, when $C/N_{0}\;>35\;\mathrm{db}-\mathrm{Hz}$, the computational load was maintained at almost the same level due to the high level of $P_{D}$, the small uncertainty M, and the low degree of $P_{F}$. However, when errors of auxiliary information increased, the degradation of $P_{D}$ and the enlarged uncertainty M affected $MAT_{Slv}$ of the typical method. Using the set up verification environment, the experimental $P_{D}$ and $P_{F}$ could be obtained. Therefore, $MAT_{Slv}$, as listed in Table 2, can be obtained to determining $T_{C}$ in the criterion (25).

On the other hand, the settling time of the primary loops, $T_{S,\;}{}_{Pri}$, were simulated to calculate the computational load of $T_{P}$ in Equation (25). $T_{S,\;}{}_{Pri}$ are listed as a look-up table that contains many loops including a typical phase locked loop (PLL), and a fuzzy tracking control loop [31]. In this study, the settling time was defined as time elapsed from a unit frequency ramp input to the time at which the output tracked results remained within the specified error bound of the lock limitation of the slave loop. We measured the error between the output frequency of the master loop with respect to the reference Doppler frequency shown on the screen of the GSS8000 simulator. The Monte Carlo simulations contain zenith satellites and different receiver dynamics with different gravitational accelerations (G), which are the specific gravitational force acting on one body in a gravitational field with a Doppler frequency ramp of 51.52 Hz/G in Table 3 [31].

In Figure 12, we used the continuous experimental probability of detection and false alarm rate that covered the received $C/N_{0}$ from 28 db-Hz to 38 db-Hz to show the complete $MAT_{slv}$ trend. By considering the following $R_{E\;}$ and $R_{RLS\;}$ analysis, Equation (25) can be computed.

#### Analysis of the Running Time ($R_{E}$ and $R_{RLS}$)

For the running time $R_{E\;}$, only 10~20 ns is needed as the conventional aid acquisition uses the estimation (3) with a simple constant of proportionality. On the other hand, the running time of the robust estimator $R_{RLS\;}$ is longer than $R_{E\;}$, affected by the dimension of the matrix in the model and the level of noise [32]. Based on experiments on the ARM cortex-A9 core using a build-in “Global Timer” [33], when the signal-to-noise ratio >25 dB-Hz, 10 iterations of (18) can converge to a relative error of 0.08% within 15~25 µs to obtain an optimized μ and ${\hat{X}}_{RLS}$. It is seen that using this FPGA platform [17], $T_{S,\;}{}_{Pri}$ and $MAT_{Pri}$ dominate the computational load, since their elapsed time of several hundred ms was dozens to hundreds of times longer than $R_{E\;}$ and $R_{RLS\;}$.

### 5.3. Analysis of the Bit-Boundary Alignment Strategy

The tracked bit boundaries maintain clear edges during low received signal-to-noise ratio. Hence, they can be used as code phase indicators for the second band. Figure 13b presents the punctual, late, and early correlation results of Figure 13a with the 0.5 chip space, and it can be seen that the timelines of Figure 13a,b are aligned. When the $C/N_{0}\;<\;30$ dB-Hz (after 0.25 s), the amplitude of correlation results from the early and late correlators (blue and yellow lines) was closer to the punctual correlator (orange).

In the low signal-to-noise ratio situation, if the code acquisition is used as an auxiliary code phase provider, it may be subject to increased uncertainty of the assist code phase. However, using the primary tracking scheme, tracked bit boundaries are not only used to quickly solve the ambiguity, but used to correctly point to the code phase of the secondary channel as the sharp tracked bit edges have high measurement sensitivity during the bit and correlated power inversion, which are not yet obtained in the acquisition stage.

### 5.4. Performance of the Robust and Adaptive Link

The robust link took the auxiliary information from the primary tracking loop to obtain the bound error and continuity of the auxiliary data. By using the obtained upper error bound for (12), the robust estimator can solve an optimized parameter to improve the robustness. Figure 14 shows the performance comparison between the typical LS and RLS, when the 3-standard deviation boundaries are obtained and given to the RLS (yellow curve), the RLS can obtain a higher level of robustness than the typical LS linker (blue curve) as the optimized μ can damp the propagated structural uncertainty.

A typical aid scheme and the robust aid scheme were compared. In the former, the assist rate was very slow due to the master acquisition, which considered a frequency resolution of 125 Hz and code step of 0.5 chip and had to spend 513 ms to complete one coarse search to provide one auxiliary data. If the error of the auxiliary Doppler frequencies exceed the initial tolerance of the slave loop (35 Hz), the direct aid tracking fails, whereas the robust aid, which uses the primary tracking and aiding scheme can better describe the channel variation. In addition, the mitigation of fixed bias $b_{f}$ was taken to make the cell search more robust toward a wide range of initial frequency offsets between two channels. Hence, as can be seen in Figure 15, the new scheme can maintain a sufficiently small initial error to directly aid the slave tracking loop. The so-call “error bias” between the generated ground truth and aided value (green line) in Figure 15 is caused by the set 125 Hz coarse frequency resolution and 0.5 chip space of the primary acquisition. If we increased the resolution to restrict the error, the computational load of the conventional aid severely diverged. Hence, the robust scheme achieved a better aiding performance in terms of the assist rate and accuracy.

### 5.5. Resource Usage of the FPGA Implementation

By using the proposed robust inter-frequency aid, the hardware resources of the GNSS FPGA can effectively be used. The established robust link can directly drive the slave loop without a slave acquisition. Therefore, some modules can be released such as the additional local numerically controlled oscillators (NCOs), which contains a look-up-table based direct digital synthesizer (DDS) and a phase generator, occupying memory resources. As presented in Table 4, compared to the typical aid scheme, the resource usage of the proposed robust aid scheme saved more than 31.02% of the static random access memory (SRAM).

## 6. Conclusions

This study aimed to establish a robust link on the FPGA to achieve better aid synchronization for a multi-carrier GNSS receiver. Typically, the aid acquisition may suffer from an error carrier injection, so the conventional primary acquisition is used to provide the assist data to the other channels, whereas this primary tracking fetches auxiliary data from a master tracking loop in the primary channel, thus, the error boundary can be propagated from the locked tracker. Then, a robust estimator was designed to directly link the two tracking loops in the primary and the second channels, ensuring robustness and the time spent on synchronization. In addition, this process uses the robust bit-boundary alignment scheme to point to the code phase in the multi-carrier signals for fast solving ambiguity.

FPGA implementation was discussed, and the experiments show that the proposed aid scheme maintained the same level of computational load when the noise was low, however, when errors between each band became severe, it maintained a smaller uncertainty, thereby preserving the computational load. Moreover, using the RLS link, the direct link of the two loops released the significant hardware resources. Therefore, the direct link significantly improved the robustness as well as the architecture efficiency.

## Figures and Tables

**Figure 1 sensors-21-04634-f001:**
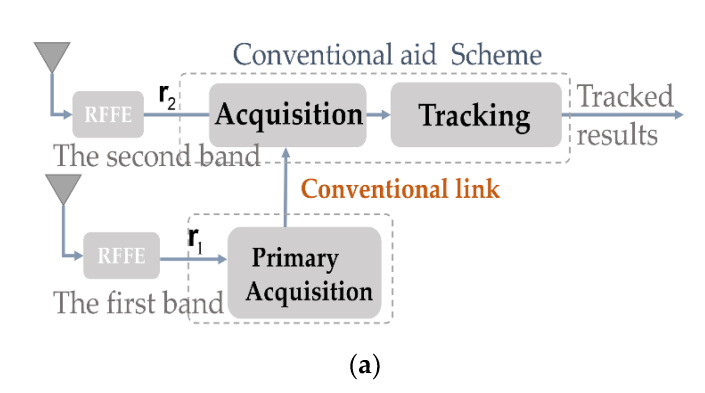
(**a**) Conventional aided acquisition. (**b**) Proposed direct aided tracking scheme and robust link.

**Figure 2 sensors-21-04634-f002:**
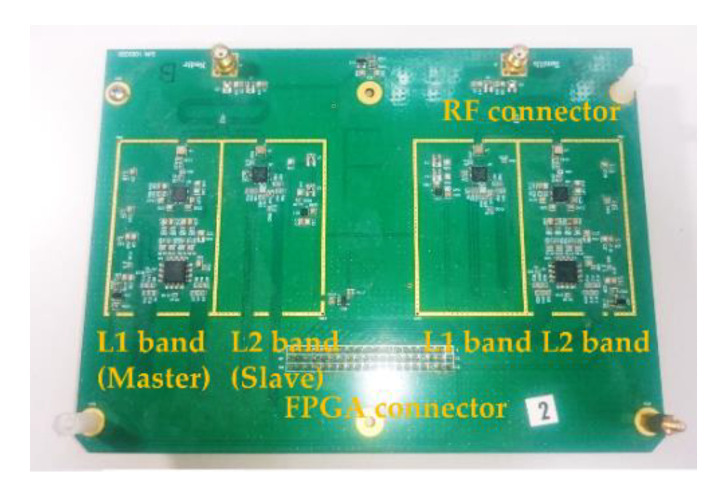
Implemented dual-frequency radio frequency front ends.

**Figure 3 sensors-21-04634-f003:**
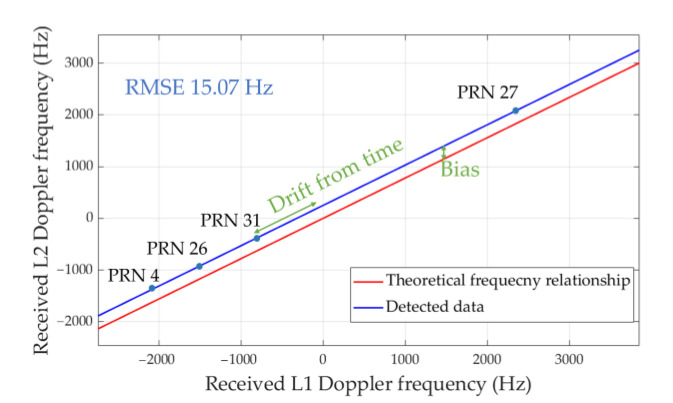
Received Doppler frequencies at two GPS bands from the experiments.

**Figure 4 sensors-21-04634-f004:**
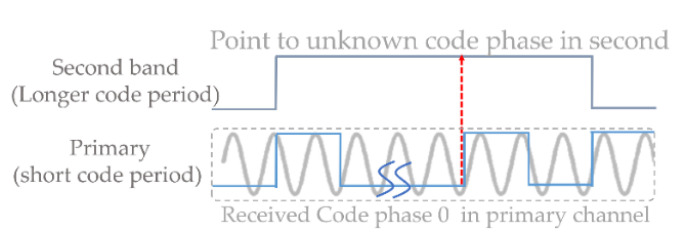
Limitation when different code periods occur in the inner code aiding scheme.

**Figure 5 sensors-21-04634-f005:**
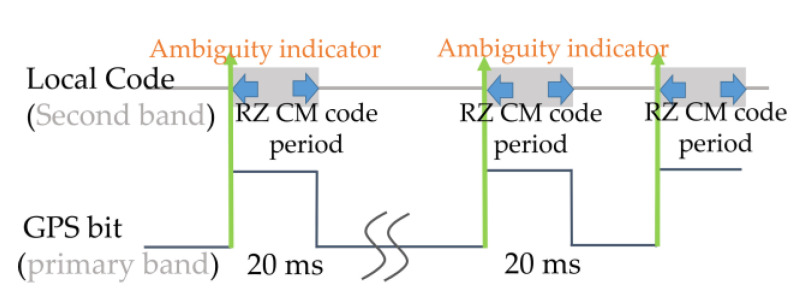
Bit-boundary assisted signal alignment.

**Figure 6 sensors-21-04634-f006:**
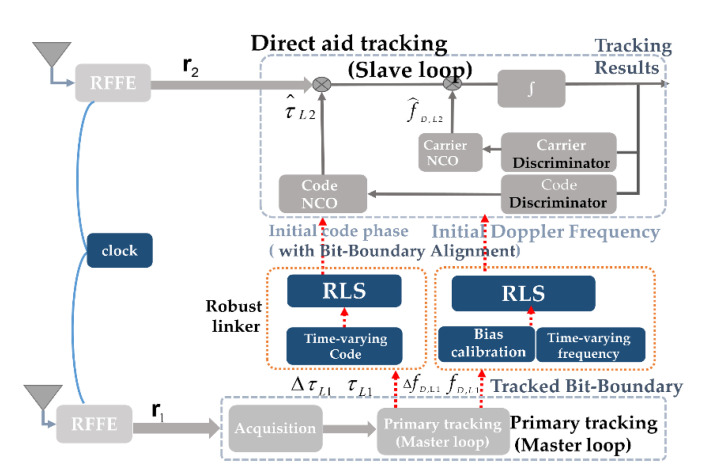
Algorithm block diagram of the proposed robust direct aid tracking scheme.

**Figure 7 sensors-21-04634-f007:**
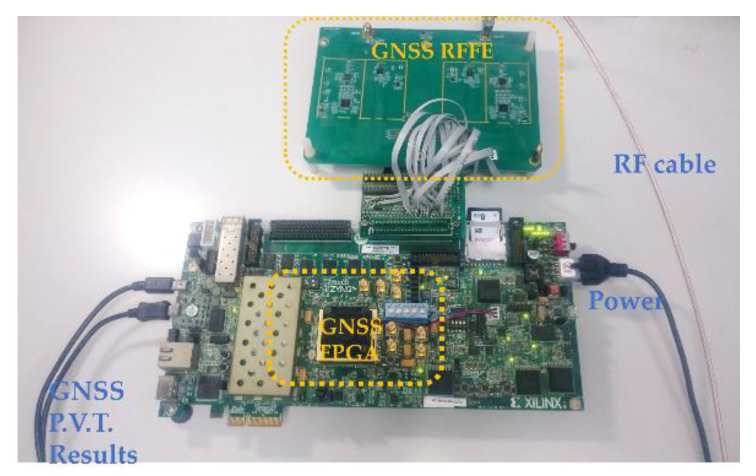
The implemented dual-frequency GNSS FPGA receiver.

**Figure 8 sensors-21-04634-f008:**
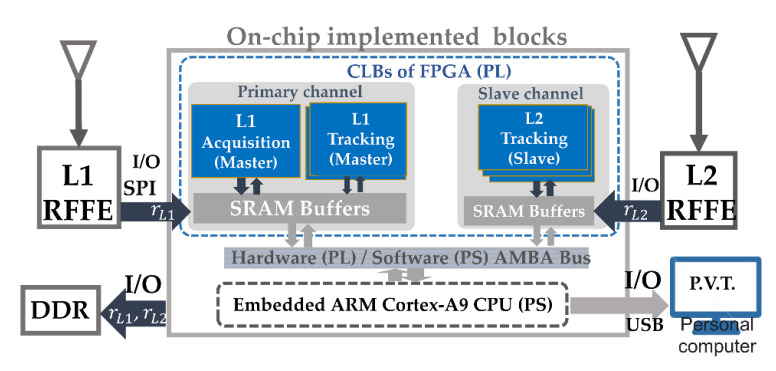
Hardware units of FPGA and implemented blocks.

**Figure 9 sensors-21-04634-f009:**
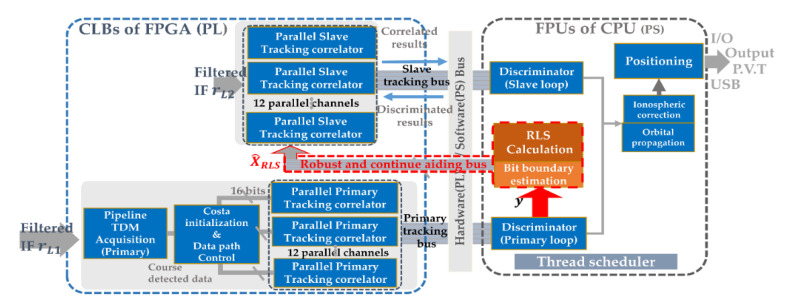
Efficient FPGA implementation with robust inner aid.

**Figure 10 sensors-21-04634-f010:**
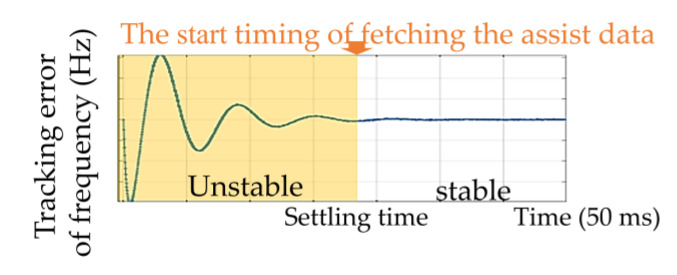
Timing for fetching data from the primary loop.

**Figure 11 sensors-21-04634-f011:**
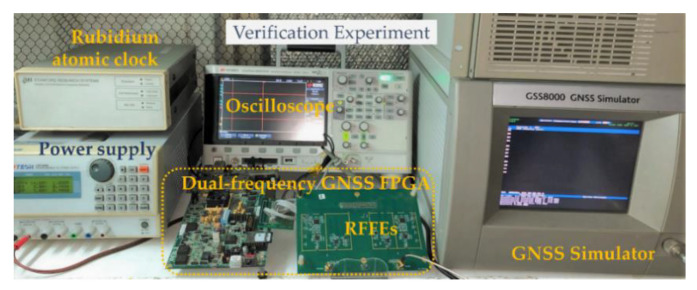
Verification environment.

**Figure 12 sensors-21-04634-f012:**
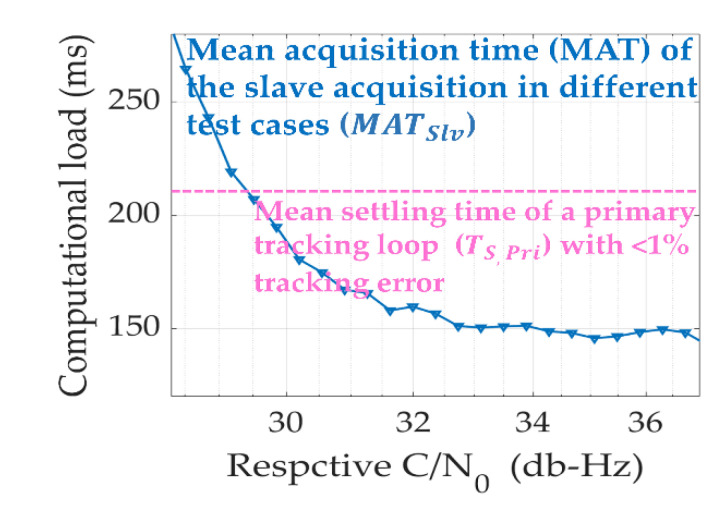
Analysis of $MAT_{slv}$ and $T_{S{,}_{}pri}$ with different $C/N_{0}$.

**Figure 13 sensors-21-04634-f013:**
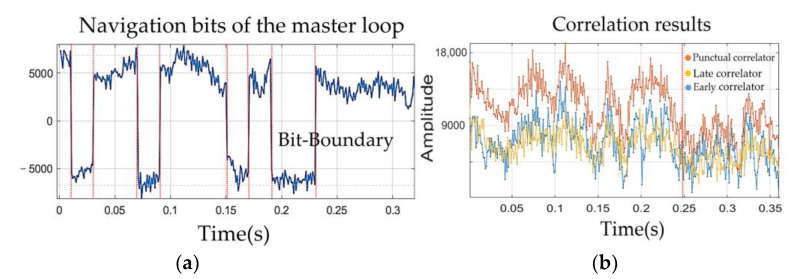
(**a**) Using tracked bit-edge as a code phase indicator for the slave loop $C/N_{0}=28$; (**b**) output of the correlator in the master loop with signal interference (0.25 s).

**Figure 14 sensors-21-04634-f014:**
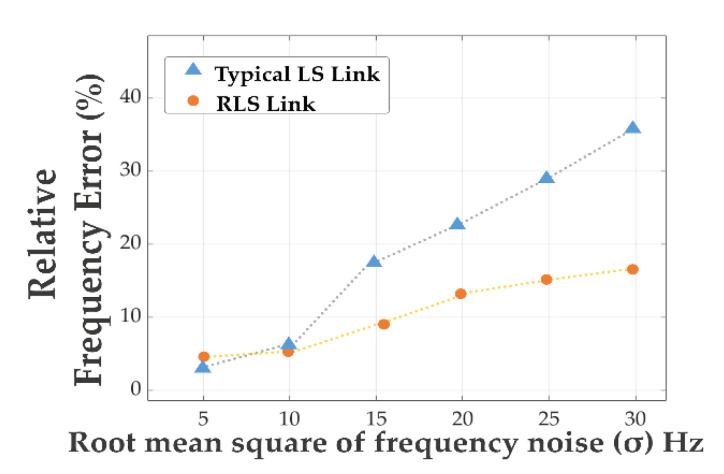
Performance of the RLS estimator.

**Figure 15 sensors-21-04634-f015:**
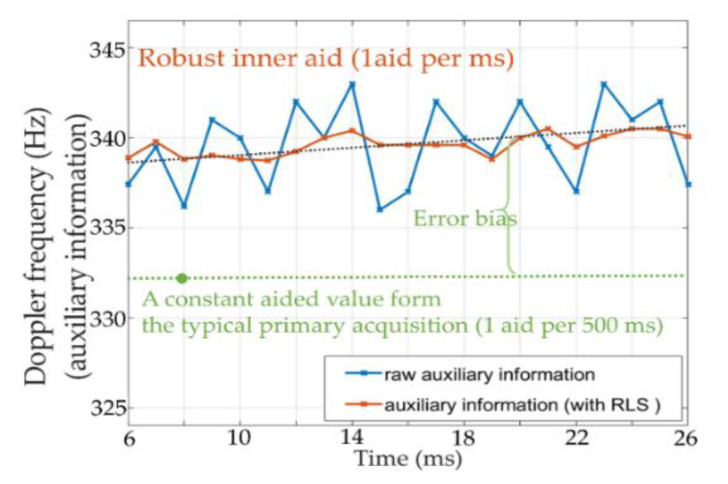
Comparison of a typical and robust aid scheme.

**Table 1 sensors-21-04634-t001:** Algorithm partition on the FPGA.

Tasks of Dual-Frequency GNSS Receiver (Baseband)	CLBs of FPGA (PL)	CPU (PS)
Fast I/F filtering and buffering (Primary and slave channels)	√	
Acquisition (primary channel)	√	
Tracking correlator (Primary and slave loops)	√	
Core bridge between FPGA logic and embedded CPU	√	√
Discriminator (Primary and slave loops)		√
Robust estimator for the inner frequency aiding		√
Pseudo range forming and positioning		√

**Table 2 sensors-21-04634-t002:** Analysis of $MAT_{slv}$ for a conventional method with different $C/N_{0}$.

	M	$P_{D}$	$P_{F}$	$MAT_{slv}\;$
Individually acquisition $C/N_{0}=35$ db-Hz	20,000HZ/100HZ×10,230×2	0.882	0.0975	5.18 h
Conventional aid $C/N_{0}=35$ db-Hz	20,000HZ/100HZ×1×3	0.895	0.1075	133.03 ms
Conventional aid $C/N_{0}=30$ db-Hz	20,000HZ/100HZ×1×3	0.701	0.213	198.81 ms
Conventional aid $C/N_{0}=28$ db-Hz	20,000HZ/100HZ×1×3	0.557	0.350	277.28 ms

**Table 3 sensors-21-04634-t003:** Settling time of the primary loop ( $T_{S{,}_{}pri}$ ) analysis.

Gravitational Acceleration (G)	1 G	2 G	3 G	4G
Frequency ramp (Hz/s)	51.5 Hz/s	102.9 Hz/s	154.5 Hz/s	206 Hz/s
Settling time (ms): 3nd Slave loop	715 ms	721 ms	730 ms	890 ms
Settling time (ms): 2rd Slave loop	228 ms	230 ms	235 ms	Lose lock
Settling time (ms): Fuzzy Slave loop [31]	7 ms	7 ms	8 ms	12ms

**Table 4 sensors-21-04634-t004:** Hardware resource utilization.

CLBs of FPGA	Typical Aided Acquisition			Primary Tracking Scheme with the Robust Inner Frequency Aiding		
	Utilization	Available	Utilization (%)	Utilization	Available	Utilization (%)
Flip-Flops	22,865	437,200	5.23%	19,805	437,200	4.53%
Look-Up Tables	18,318	218,600	8.38%	6952	218,600	3.18%
I/O	51	362	14.09%	51.01	362	14.09%
Block RAM (Mb)	15.03	19.2	78.31%	9.07	19.2	47.28%

## Data Availability

Data available in a publicly accessible repository that does not issue DOIs publicly available datasets were analyzed in this study. This data can be found in Appendix A. The collected raw data that supports the scheme of this study have been included and uploaded as Appendix A with description files and the corresponding code. The raw data collected from front-end shows real phenomenon of parameter uncertainty from environments in two bands. The video of the collection process was also uploaded.

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
