# Peer review of "Efficient FPGA Implementation of a Dual-Frequency GNSS Receiver with Robust Inter-Frequency Aiding"

_sensors, 2021, doi:10.3390/s21144634_

Round 1
Reviewer 1 Report
A. General
This paper proposes a robust aiding scheme using FPGA in dual-frequency GNSS receivers. When the primary tracking loop has been locked, the code delay and the Doppler frequency are converted appropriately using a robust estimator. The estimated values are given to the slave tracking loop as an initial value. With the proposed method, because the slave channel does not have to perform its own acquisition, the resources of FPGA can be saved.
B. Major comments
- In the analysis on the time spent for synchronization, the authors compared the computational load of conventional method and the proposed primary aiding method. However, the explanation or the result of the computational load for robust estimation process was not included. I think it would be better if this is covered in the paper.
C. Minor comments
- Across the paper, authors use the terminology ‘course’ for rough estimation such as an acquisition. I think it should be corrected as ‘coarse’ to prevent confusion.
- In line 172, there is explanation of the speed of light, but the speed of light does not exist in equation 4.
- In line 194, can you add explanations of how the value 40.3 has been obtained?
- In equation 6, I think a parameter related to solved ambiguity should be contained in the equation. Currently from equation 6, it seems like the ambiguity is fixed to 1, even though the actual ambiguity is solved to have a certain value between 1 and 20.
- In figure 6, maybe there is typing mistake. (Bit-Boundary Aliment) -> (Bit-Boundary Alignment)
- In equation 7, f_L/f_L in the parenthesis should be corrected.
- In line 258, the index number of chapter should be corrected. (3.1.3) -> (3.1.2)
- In line 358, the explanation of sigmal_(T, 1) is not exist.
- In line 368, the index number of chapter should be corrected. (3.1) -> (4.1)
- In equation 25, it seems like there is an ambiguity that which one is primary tracking scheme or typical method. Modifying the equation to have more clarity would be helpful for readers.
- In line 481, the terminology MAT_second would be better if it is unified as MAT_slave.
- In table 2, it is strange that the individual acquisition method spends as long as 5.18 hours. Please check the value.
- In line 488, the value and unit of Doppler frequency ramp (52.52 Hz/G) do not match with table 3. Please check it.
- In figure 12, the settling time of a primary loop (pink line) has a value of about 190 ms. Can you explain how this value is obtained?
- There is no explanation of figure 13(b). Please add explanation.
- Please give the information of unit for x-axis of figure 14.
- In table 4, some utilization values are not correct. Please check them.
Reviewer 2 Report
The authors present an approach for designing and implementing a dual-frequency GNSS receiver. Overall the paper is very well written and structured. There are two major issues that this reviewer has with the paper in its current state, which are:
- It is unclear where the presented work rests with respect to the current state of the art. This may be due to the fact that there is not a clear presentation/explanation of the presented work versus prior published works. The authors need to clearly describe how their work compares and goes beyond currently published works.
- The other is a formatting issue. Many of the large equations and pretty much all of the figures extend into the left margin of the page. These 'overhangs' need to be corrected.
In addition to these major issues, this reviewer has one additional minor suggested change. That is the removal of words as variables and subscripts to variables in mathematical expressions. One example is the use of 'ambiguity' in equation (5). Variables need to be a single letter or character and subscripts need to be short as well (i.e., one to two letters or characters). This reviewer is okay with the subscripts that are comma separated such as in equation (4). However, the use of 'resist' in equations (20) and (21) is not an acceptable case. Please correct these as well.
Reviewer 3 Report
This is an important study demonstrating a new efficient approach to synchronous detection of the GNSS signals in L1 and L2 bands. Overall, the quality of the presentation is good, the concept is well supported by the experimental data. Please check carefully the English language through the text, format equations and figures to be inline with the text and check the format of the references
Round 2
Reviewer 2 Report
Dear Authors,
Thank you for addressing all of my comments within your revised manuscript. I have no other comments, questions, nor concerns at this time.
This manuscript is a resubmission of an earlier submission. The following is a list of the peer review reports and author responses from that submission.
Round 1
Reviewer 1 Report
Summary
This paper provides a new robust link in the aid synchronization and signal combination for a multi-carrier GNSS with resource efficiency. The proposed method could maintain the signal combination performance when C/N0 is low. The proposed method deserves further study, and followings are some detailed comments.
Major comments
1.The conclusion only describes the advantages of the proposed method, and its shortcomings and limitations should be supplemented.
2.The conclusion provides an overview of the content of the paper. In addition, the broad application prospects of the proposed method should also be briefly explained, so that researchers are interested in this method.
3.More favorable experiments in different C/N0 are needed to enrich the contents of Table 1 and Table 2. Besides, Timing performance should be compared in different scenarios to present the experimental results more clearly. eg: Line chart is an efficient way.
4.The comparison dimension of robustness should be more diversified. In addition to comparing the timing performance, it is also advisable to compare the carrier phase error or positioning error obtained by the proposed method and the traditional method.
Minor comments:
- The quality of the figures in the paper needs to be improved.
- There are spelling errors in some pictures. eg: The Abbreviation of Radio Frequency Front End is RFFE instead of RFEE in Fig1,2 and 7.
- The elements in some pictures are not arranged neatly, its size is unreasonable so that readers cannot see the content clearly (Fig5, 7).
- The text in some pictures is not clear (Fig2)
- Some paragraphs are not indented.
- A pdf file named "Advice" which notes some mistakes was attached below.

Reviewer 2 Report
[General Comments]
This manuscript describes a new robust method to treat multi-carrier GNSS signals. The Doppler frequencies of L1, L2 and L5 bands are no longer limited to proportional relationship but generalized in this manuscript. Generally, I do appreciate ambitious challenging researches, but I’m afraid this manuscript is not in a complete form; for example, no descriptions for Figure 15, Tables A1 and A2 are found in the manuscript, although they seem to provide important results. Consequently, I am not fully convinced with the manuscript since the results of the methods are not well described, although the authors have spent much efforts to describe the methods.
Although the authors emphasize qualitative advantages of the new method, many readers would prefer to know quantitative comparisons with the traditional methods. Does the new method improve the estimation accuracy of Doppler frequencies themselves, or does it simply aim to reduce computational resource? Please describe how they have been qualitatively improved from the traditional methods (I guess as in Fig 15), and whether these differences are practically significant or trivial.
Therefore, I would conclude that this manuscript needs major revision, especially in the later half of the manuscript where describes how the new methods improve the traditional results.
[Additional Comments]
P5 Fig3, P8 Eq (7) Since this is a key concept of this manuscript, it would be helpful for general readers to explain why the Doppler frequencies may be distorted from the ideal proportional relationship. Unless, readers would have no idea whether these distortions are commonly observed or seldom occur.
P7 Eq (6) The carrier power “P” is treated as a constant in Eq (6), but the actual power received by an antenna would fluctuate in time by many environmental effects. Does these fluctuations included in “observed Doppler frequency” f_D (as a source of the Doppler relationship distortion)?
P17 Table2 Where can I find description on “Gravitational acceleration”?
P17,18 Figs 13 and 14 Put values of the ordinate axis in Figures 13b and 14a.
P7 L265 Is the second P_L1 typo of P_L2?
Round 2
Reviewer 2 Report
I found the authors have extensively revised the manuscript. I am satisfied with the revision so recommend it to be published.
P.S. Line numbers 804 and 805 in Table A2 are distorted.
